# *Lacticaseibacillus rhamnosus* GG in a chewable colonizes the nose and facilitates local immune benefits in allergic rhinoconjunctivitis patients

Ilke De Boeck,[1] Irina Spacova,[1] Eline Cauwenberghs,[1] Tom Eilers,[1] Thies Gehrmann,[1] Karlien Van den Bossche,[2,3] Katleen Martens,[4] Sandra Condori-Catachura,[1] Kato Michiels,[1,5] Fien De Winter,[6] Samir Kumar-Singh,[6] Nicolas Bruffaerts,[7] Ann Packeu,[7] Peter W. Hellings,[4,8] Anneclaire Vroegop,[2,3] Klara Van Gool,[9] Olivier M. Vanderveken,[2,3] Sarah Lebeer[1]

**ABSTRACT** Current treatments fall short in managing allergic rhinitis (AR), emphasizing the need for additional strategies. Beneficial bacteria application shows promise in AR; however, most studies focus on oral probiotic administration without monitoring the applied strains in the upper respiratory tract (URT) and their local effects. In this randomized, double-blind, placebo-controlled trial, the probiotic *Lacticaseibacillus rhamnosus* GG was administered via chewable tablets in seasonal AR patients, randomized to probiotic ($n = 33$) or placebo ($n = 31$) groups. Per-protocol analysis of the URT microbiome, immune markers, and AR symptoms was performed. *L. rhamnosus* GG trafficked from chewables to the oropharynx (77%, $P = 0.02$) and nasopharynx (41%, $P < 0.0001$). Control of self-reported AR symptoms via validated questionnaires under grass pollen exposure was observed after 2 weeks of probiotic administration and not upon placebo. A local decrease in salivary interleukin-4 ($P < 0.05$) and nasal IL-13 ($P < 0.0001$) was observed in the probiotic group. These data indicate that *L. rhamnosus* GG chewables can target the URT and exert local effects on key allergy cytokines after temporal probiotic engraftment.

**IMPORTANCE** Allergic rhinitis (AR) or hay fever is a highly prevalent condition, impacting nearly half the population in some countries. Supplementation of beneficial bacteria or probiotics has gained increasing attention in AR, and a key innovative way to do this is direct administration to the upper airways. Our study shows for the first time that the model probiotic strain *Lacticaseibacillus rhamnosus* GG can traffic to the nose in AR patients when administered via a slow-releasing chewable tablet. This trafficking is associated with local benefits in the airways, including on grass pollen-induced nasal symptoms and allergy-related cytokines.

**KEYWORDS** human microbiome, probiotics, allergy, *Lacticaseibacillus*

In a world where modern living increasingly distances us from natural environments, the groundbreaking work on the Old Friends hypothesis (1) and the biodiversity hypothesis (2) converge and both point toward the trend that reduced contact with natural environments leads to disruptions in the delicate balance of our microbiome and immune system functioning. This contributes to a global increase in allergic and inflammatory disorders. Notably, one such disorder is seasonal allergic rhinitis (AR) or hay fever, a prevalent condition that disproportionately affects high-income industrialized countries, impacting nearly half the population in some countries. The implications are vast, with a serious impact on the patient's quality of life (QOL) and a significant economic and health burden (3).

Address correspondence to Sarah Lebeer, sarah.lebeer@uantwerpen.be.

Ilke De Boeck and Irina Spacova contributed equally to this article. Author order was determined based on clinical study expertise.

This project was partially funded by DSM I-health. They also provided the study probiotic and placebo. This sponsor was not involved in the original study idea brainstorms, study design, data acquisition, data-analysis, and interpretation of the results. S.L. also reports research funding and collaborations with other industrial partners unrelated to this work.

See the funding table on p. 14.

It is now well established that the pathogenesis of AR involves immune imbalance with a key role for interleukin (IL)-4, causing a Th2 immune response and IgE-mediated allergy (4). In addition, IL-4 as a master Th2 cytokine drives the generation of pro-allergic IL-5 and IL-13 (5). Current systemic (e.g., antihistamines) and local AR drug treatment or prevention strategies (e.g., intranasal corticosteroid sprays) focus on symptom relief and reduction of the inflammatory response, but they do not address the underlying immune imbalance linked to AR (6, 7). For many patients, symptoms are not adequately controlled with the current strategies (approximately 20% in patients with severe and persistent symptoms), and there are often side effects, illustrating the need for alternative and/or add-on treatment options (8).

In addition to the immune imbalances, we and others propose that new treatment strategies could originate from the new insights in the microbiome (reviewed in references 9, 10). Indeed, specific local microbiome imbalances have been described in nasal samples of AR patients based on sequencing studies, such as an increase of *Streptococcus* taxa (6, 11) and a decrease of beneficial lactobacilli (12). We have also recently found that lactobacilli, including *Lacticaseibacillus* species, are common low-abundant members of the human upper respiratory tract (URT) (13). They were reduced in prevalence and relative abundance in patients with chronic rhinosinusitis (CRS), an airway disease that has several features in common with AR. In an *ex vivo* model, a specific *Lacticaseibacillus* strain was also able to restore airway epithelial integrity in explants obtained from CRS patients with nasal polyps (14). These findings stimulate the exploration of the nasal application of lactobacilli. However, direct nasal applications with live probiotics have to follow the guidelines for live biotherapeutic products (LBPs), which is a new and yet unclear regulatory path for drugs with live microorganisms such as probiotics (15). Therefore, oral application with selected probiotic strains is still the preferred administration route. A recent meta-analysis has already shown the clinical benefits of oral intervention with selected probiotics for AR (16).

The model probiotic strain *Lacticaseibacillus rhamnosus* GG has been the focus of many allergy-related clinical research trials, demonstrating the potential of this strain to exert systemic immunomodulatory effects when applied orally, via the gut-lung axis (see (17) for an overview). For example, *L. rhamnosus* GG has previously been shown to prevent the development of early atopic disease in children at high risk when administered orally to mothers and infants during the first six months of their life (18), and this preventive effect extended even up to the age of four years (19). However, the available double-blind, placebo-controlled studies do not uniformly point to clinical efficacy in populations with AR. For instance, preventive administration of oral capsules of *L. rhamnosus* GG in 38 birch-pollen sensitive teenagers and young adults before and during the birch pollen season did not significantly alleviate symptoms or reduce medication scores when compared to placebo capsules containing microcrystalline cellulose (20). On the other hand, a slightly larger trial in younger children ($n = 100$; age 5–12 y/o) with *L. rhamnosus* GG as add-on therapy to sublingual immunotherapy showed an enhanced immune response in the group with probiotics compared to the immunotherapy alone (21). Also, others have used *L. rhamnosus* GG as an add-on to corticosteroids in persistent adult AR patients and found significant improvement in the QOL of the patients (22). When *L. rhamnosus* GG was combined with *Lactobacillus gasseri* TMC0356 and consumed as fermented milk before and during pollen season, this could alleviate symptom scores for nasal blockage and medication scores in 44 adult patients with seasonal AR caused by Japanese cedar pollinosis (23). Taken together, these data suggest that the specific timing of intervention, administration regimen, patient population, and probiotic formulation (e.g., oral capsule and fermented drinks) have a large impact on the clinical efficacy, even for the same probiotic strain such as *L. rhamnosus* GG. In addition, whether the URT is—in addition to the gut—also a direct site of action is for most studies not known. To the best of our knowledge, no study executed so far with *L. rhamnosus* GG, when applied orally, has explored whether the strain can migrate to the

nose. Research in mouse models suggests that nasal administration of *L. rhamnosus* GG could be more effective compared to oral administration, for instance, for the prevention of the development of birch pollen-induced allergic asthma (24).

In this study, we aimed to explore whether the model probiotic *L. rhamnosus* GG could traffic to the nose when applied in a formulation that promotes prolonged release in the oronasopharynx and whether this prolonged contact with the nasal mucosa could result in local benefits in the URT. Hereto, we designed a double-blind, placebo-controlled, randomized trial in patients with seasonal AR (*n* = 64) with *L. rhamnosus* GG administered in a commercially available slow-release chewable and this in a semi-preventive set-up before the grass pollen season started. The chewable formulations were hypothesized to promote the local contact with the oral mucosa during chewing compared to standard capsules that are immediately ingested. In addition, we evaluated the effects of this probiotic intervention on total nasal symptoms scores of patients with AR in relation to their exposure to airborne pollen, local cytokine modulation, and microbiome composition of the oropharynx and nasopharynx.

## RESULTS

### Set-up of a placebo-controlled intervention trial in patients with AR during grass pollen season in adjuvant setting

Eighty-seven participants were assessed for eligibility, of which 64 were randomized (33 to the probiotic treatment and 31 to the placebo treatment). Most participants who were not eligible were excluded because of cross-allergy with other allergens (mainly house dust mite) or not having a reaction against grass pollen based on skin prick test (SPT). Fig. 1A depicts patient recruitment and enrollment. Three participants dropped out during the trial (two probiotic, one placebo; reasons in Fig. 1A) and per protocol analysis of the remaining participants that provided samples on all time points was conducted. The study set-up and actual pollen concentrations, monitored in the aerobiological station in Brussels, in relation to the start dates of this study are shown in Fig. 1B. Patient demographics and baseline characteristics are shown in Table 1. Patients were allowed to use rescue medication (see Table 2 for baseline levels and Table S1 for medication use at start and during the study). None of the participants showed problems with general oronasopharyngeal health, as assessed by the responsible clinician at each visit.

### *L. rhamnosus* GG traffics to the oronasopharyngeal region after administration in a chewable tablet

The primary objective of this trial was to evaluate whether orally applied *L. rhamnosus* GG in a chewable was able to transfer to the oronasopharyngeal region to exert local effects. Trafficking was evaluated via qRT-PCR with strain-specific primers developed in-house for *L. rhamnosus* GG. After the intervention period of 8 weeks, 13 out of 31 (42%) and 24 out of 31 (77%) participants in the probiotic group had detectable ($>10^2$ CFUs) *L. rhamnosus* GG via qPCR for nasopharynx and oropharynx, respectively, while *L. rhamnosus* GG was not detected at start (Fig. 2A). The 11 out of 13 participants who had detectable *L. rhamnosus* GG in the nasopharynx were also positive for oropharynx (paired samples are indicated with a gray line in Fig. 2A). In the placebo group, one participant showed detectable ($>10^2$ CFUs) *L. rhamnosus* GG in his/her nasopharynx at the start. After an 8-week intervention, 5 out of 30 participants (16.6%) and 4 out of 30 (13.3%) participants were also positive for the nasopharynx and oropharynx. The detection levels at the end of the trial showed that *L. rhamnosus* GG was detected with significantly higher amounts in the probiotic group compared to placebo, with $P = 0.02$ and $P < 0.0001$ for nasopharynx and oropharynx, respectively (Mann-Whitney test).

In addition to the qPCR analysis with strain-specific primers, 16S rRNA amplicon sequencing of the V4 region revealed a strong correlation between the *L. rhamnosus* GG amplicon sequence variant (ASV) and the probiotic treatment group, both for the

**TABLE 1** Patient demographics and baseline characteristics by treatment group of participants that finalized the study[a]

|  | Probiotic (n = 31) | Placebo (n = 30) |
|---|---|---|
| Age (years) [mean, stdv] | 36 ± 8 | 38 ± 10 |
| Sex (female) [n, %] | 13 [42%] | 16 [53%] |
| Smoker (yes) [n, %] | 2 [6.5%] | 0 [0%] |
| Lung disease (asthma, COPD) | 5 [16%] | 3 [10%] |
| Immune disorder (yes) [n, %] | 0 [0%] | 0 [0%] |
| Diabetes (yes) [n, %] | 0 [0%] | 0 [0%] |
| Hypertension [n, %] | 0 [0%] | 0 [0%] |
| TNSS at start [mean] | 1.64 | 1.59 |
| SNOT-22 at start [mean] | 17.2 | 17.7 |
| Intermittent, mild [n, %] | 2 [6.5%] | 3 [10%] |
| Intermittent, moderate/severe [n, %] | 5 [16%] | 3 [10%] |
| Persistent, mild [n, %] | 2 [6.5%] | 1 [3%] |
| Persistent, moderate/severe [n, %] | 22 [71%] | 23 [77%] |
| Antihistamine use at start [n, %] | 25 [80.6%] | 24 [80%] |
| Corticosteroid spray use at start [n, %] | 14 [46.7%] | 7 [23.3%] |

[a]TNSS = total nasal symptom score; SNOT-22 = sino-nasal outcome test.

nasopharynx and oropharynx (Fig. 2B). The correlation with the oropharynx was stronger than for the nasopharynx, in line with the qPCR data analysis. To evaluate whether the probiotic intervention caused a shift in the microbiome profiles before and after the intervention, the diversity of taxa before and after treatment in both URT niches was investigated using the Bray-Curtis dissimilarity and visualized with principal coordinate analysis (PCoA), followed by an adonis permutation test (25). A clear distinction between nasopharynx and oropharynx microbiome profiles was observed (Fig. S1 through S3), and location determined 46% of variation in our data set ($P < 0.001$). Some shifts were detected for nasopharynx (Fig. 2C) and oropharynx (Fig. 2D) at the end of the study within participants, but these shifts seemed independent of the intervention, as they occurred in both probiotic and placebo group (not significant).

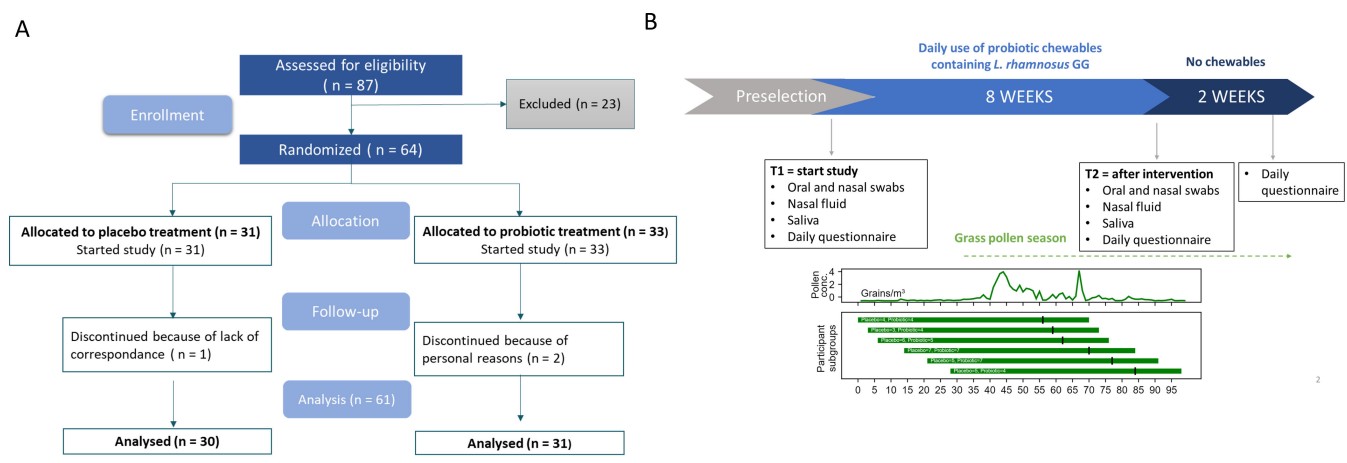

**FIG 1** Study overview. (A) CONSORT flowchart for patient recruitment and enrollment. Participants that were excluded did not meet the inclusion criteria (mostly because of cross allergy with house dust mite or not having a reaction on grass pollen based on skin prick test [SPT]). (B) Schematic overview of the study set-up and monitoring of grass pollen concentrations (grains/m³). Participants started on different start dates, indicated by the green horizontal lines for the different start groups with the size of each subgroup shown as well in relation to the environmental pollen concentrations throughout the study. Black vertical lines indicate when participants stopped taking the chewables, followed by a 2-week follow-up.

**TABLE 2** Primers used for qPCR

| Species | Primer | Sequence (5′–3′) | Reference |
|---|---|---|---|
| *L. rhamnosus* | LGG_1964_F | CTGGCACTCATGAATCCTTACA | This study |
| *L. rhamnosus* | LGG_1964_R | CCATTCGGTAGGCTACTTCTTC | This study |

## Impact of the probiotic chewables on self-reported AR symptoms via validated questionnaires

We then aimed to explore the impact of the probiotic intervention on self-reported AR symptoms using validated questionnaires and documented the total nasal symptom score (TNSS) over time per participant per treatment group. We visualized these data under the timeline of the outdoor exposure to airborne grass pollen for the entire study period (8 weeks intervention + 2 weeks of follow-up) (Fig. 3). The severity of TNSS aligns with grass pollen concentrations measured in the air per study day (average correlation per participant: 0.49 ± 0.18).

To investigate whether the probiotic chewable-treated group could control symptoms better than the placebo group, a mixed-effects linear model was used. This model considered the intrapersonal effect of each participant over the study period with multiple time-point measurements. Treatment, pollen concentrations, the use of rescue medication, and history of corticosteroid use, and their interactions were applied in the model. In addition, to account for the fact that a probiotic mode of action requires time and can result in a minimum time needed to treat, we also evaluated the effect size when removing days one by one (depicted as threshold on the *x*-axis). When applying the model to the entire study group, no significant results were observed, although trends toward lower TNSS in the probiotic group were found. To further look into this trend, the same analysis was done, but we only added participants of the probiotic group where *L. rhamnosus* GG was detected based on the qPCR results (LGG responders, 26/31 participants) and removed participants in the placebo group that had detectable *L. rhamnosus* GG (9/30) because of uncertainty of the origin of these positive and negative results (e.g., technical error and lab contamination) (Fig. 4A through D) Independent of the treatment, increased pollen concentrations resulted in significantly higher TNSS ($P < 0.05$), as indicated by their positive effect sizes (Fig. 4A). When looking at the effect of probiotic treatment, participants in the probiotic group had lower TNSS than participants in the placebo group, as illustrated by the negative effect sizes significant from days 21 to 34 ($P < 0.05$) (Fig. 4B). Our model also allowed us to explore when the probiotic intervention showed most additional clinical benefit. Around 15 days, a tipping point was observed where the effect size showed a drop, indicating a time needed to treat of circa two weeks for this probiotic intervention (Fig. 4B). The strongest effect was observed 29 days after the start of the study (depicted in Fig. 4 as black dot). Rescue medication had a significant effect in the first days of the study, but after day 12 did not have a significant effect (Fig. 4C), coinciding with a larger protective effect size of the probiotic treatment (Fig. 4B). Overall, participants in the probiotic group exhibited lower TNSS over the range of pollen concentrations (Fig. 4D), though the protective effect was more limited when participants were also taking rescue medication. Finally, we evaluated whether the presence of the *L. rhamnosus* ASV in the microbiome profiles of the oro- and nasopharynx was associated with TNSS, but no effect was observed (Fig. 2B). However, TNSS was positively associated with alpha diversity (Fig. 2B).

## Local immunomodulatory properties of the *L. rhamnosus* GG chewables and standard treatment of care in AR patients with hay fever

We next investigated the local immunomodulatory capacity of *L. rhamnosus* GG chewables. Th1 and Th2 cytokines were measured in saliva and nasal fluid obtained from the participants at baseline (start) and at week 8 (end of the treatment). We specifically focused on IL-4, as the main marker of the Th2 immune response and IgE-mediated disease and monitored IL-5 and IL-13 as pro-allergic cytokines. The IL-4 levels were

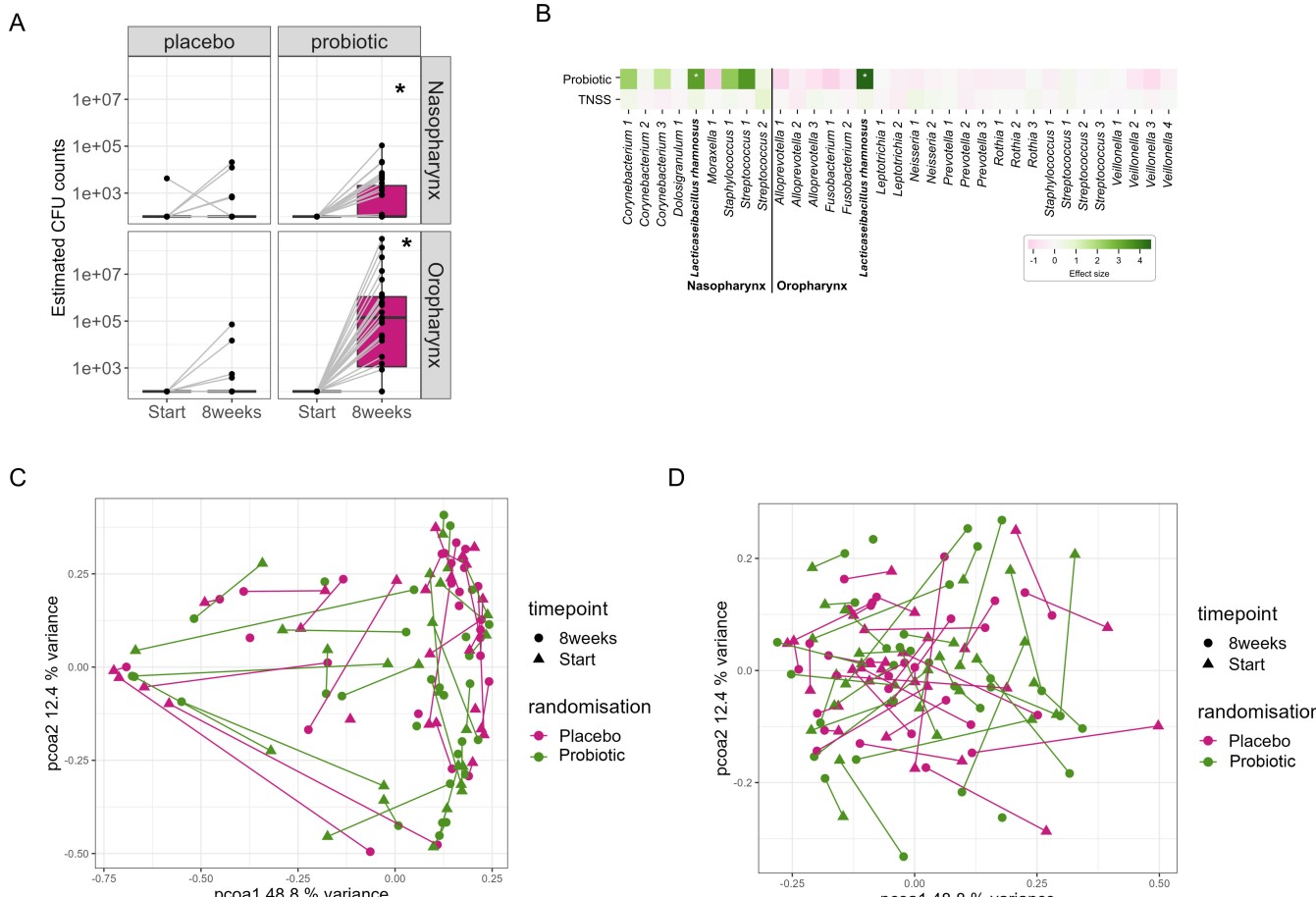

**FIG 2** Detection of *L. rhamnosus* GG in the oronasopharyngeal region. (A) qRT-PCR with strain-specific primers in the probiotic group. The detection limit was estimated to be at $10^2$ CFUs. Paired samples from individuals are shown with a gray line. (B) Association of probiotic treatment and TNSS with bacterial taxa based on 16S amplicon sequencing of nasopharynx and oropharynx samples. Effect sizes are shown ranging from dark pink (negative effect size) to dark green (positive effect size). Significant results between placebo and probiotic treatment are indicated with an asterisk (*). (C) Principal coordinate analysis (PCoA) to visualize the diversity of taxa before (start, triangles) and after (8 weeks, dots) the intervention in the nasopharynx. (D) PCoA to visualize the diversity of taxa before (start, triangles) and after (8 weeks, dots) the intervention in the oropharynx.

significantly decreased ($P < 0.05$) in the saliva of the probiotic group after 8 weeks of intervention compared to baseline (Šídák's multiple comparisons test), while this was not significant in the placebo group (Fig. 5D). Furthermore, a significant decrease ($P < 0.0001$) in nasal fluid IL-13 levels was observed in both the probiotic and the placebo groups from baseline to week 8 (Fig. 5C). Two-way ANOVA analysis identified time (start or end of study) as a significant source of variation for the levels of IL-4 in saliva and IL-13 in nasal fluid, while individual differences between participants were a significant source of variation for IL-4, IL-5, and IL-13 in saliva, and IL-13 in nasal fluid. For the other measured cytokines, no significant beneficial changes in cytokine levels were observed in the probiotic treatment group as compared to placebo.

## DISCUSSION

In this study, we evaluated whether orally administered *L. rhamnosus* GG in a chewable could traffic to the nose and induce local immunomodulatory effects in patients suffering from hay fever. We evaluated short-term persistence of the administered probiotic in the nose through a combination of microbiome analysis and strain-specific monitoring via qPCR. In addition, we assessed the local effects of *L. rhamnosus* GG chewables in the nose and throat on key allergy cytokines.

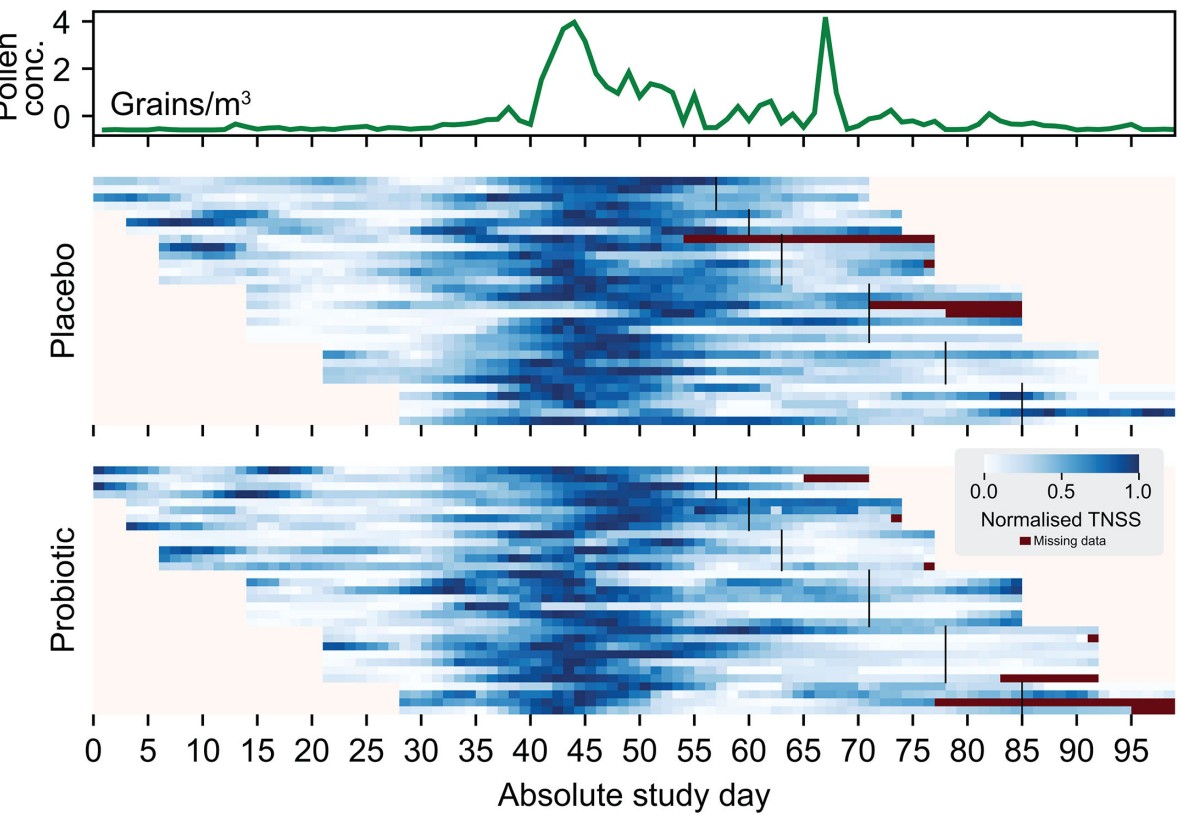

**FIG 3** Airborne grass pollen concentrations (grains/m³) aligned with TNSS per day per participant. Each horizontal line in the blue color scale represents one participant and their start date (of note, not all participants started on the same day, depicted as absolute day on the x-axis of the figure). TNSS severity was depicted from white to dark blue per day (within each participant, 0 and 1 refer to the lowest and highest reported TNSS, respectively, see scale); dark red is missing data that is non-imputed. Black vertical lines indicate when participants stopped taking the chewables, followed by a 2-week follow-up.

Our analyses provide strong evidence for trafficking and temporary engraftment in the URT after oral administration of a probiotic containing chewable. Strain-specific primers confirmed the presence of *L. rhamnosus* GG in 42% of the nasopharynx and 77% of the oropharynx samples of our study group after the study period. A strong association of the probiotic-treated group with the *L. rhamnosus* ASV was also observed in the 16S amplicon sequencing data for both respiratory niches at the end of the intervention. Of note, after the 8-week intervention group, some of the participants in the control group were also positive for *L. rhamnosus* GG via qPCR, but this was significantly lower compared to the probiotic group ($P = 0.02$ for nasopharynx and $P < 0.0001$ for oropharynx). Possible explanations include its presence as a low-abundance member of the normal nasal microbiota, as previously reported (13). Additionally, transfer through the consumption of other fermented or dairy products cannot be excluded. Finally, the potential for low-level cross-contamination during sample processing or sequencing should also be considered. To the best of our knowledge, our study is the first to report trafficking of *L. rhamnosus* GG to the oronasopharyngeal region after consumption via a chewable tablet. These results confirm our hypothesis that such chewables increase the retention of the probiotics in the mouth and in this way enable transfer of the probiotics throughout the URT for more local effects. To date, only one other randomized, placebo-controlled trial has been conducted with URT administration of potential probiotic strains in seasonal AR patients (26). In this study (26), patients ($n = 24$) were administered a probiotic mixture containing *Lactobacillus rhamnosus* SP1, *Lactobacillus paracasei* 101/37, and *Lactococcus lactis* L1A in the nostrils daily for 3 weeks in a nasal allergen challenge model. No evidence of persistent colonization was observed in nasal samples using MALDI-TOF after a treatment-free interval of 2 weeks. The authors did not

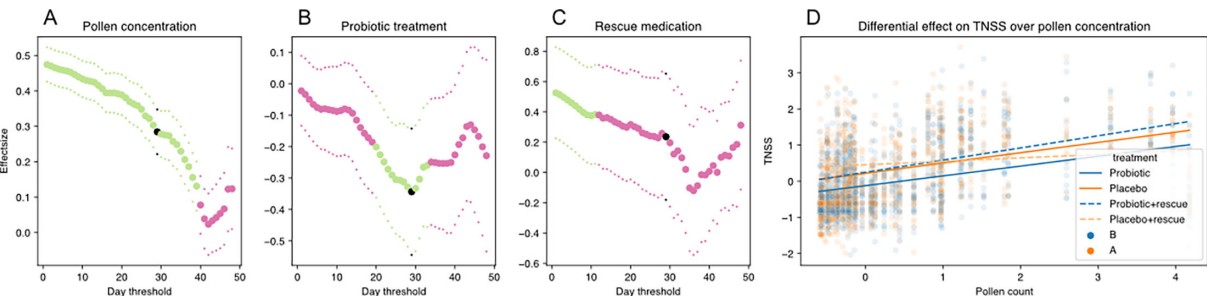

**FIG 4** Impact of treatment on TNSS in relation to grass pollen exposure and medication use in terms of effect size. (A–D) Mixed effect linear model considering treatment, pollen concentrations, rescue medication intake, and their interactions to investigate the effect on the TNSS. When considering the first *x* days as a minimum time to treat period (plotted on the *x*-axis), the effect sizes from (A) pollen concentration (POAC; family Poaceae, grasses), (B) probiotic treatment, and (C) rescue medication are plotted on the *y*-axis. Significant values are plotted in green, insignificant values in pink. Negative effect size reflects a lower TNSS. The black points indicate the selected time point where the effect size is at its maximum of the model, which is an indication for the minimal time to treat. (D) Linear regression at day 29 threshold, which has maximum effect for probiotic treatment, showing the interaction of TNSS and pollen counts for both groups with (dotted lines) and without (fixed lines) rescue medication.

report on the engraftment of the applied strains during or immediately after the study period.

In our study, daily monitoring of TNSS via self-reporting also indicated that the *L. rhamnosus* GG chewables could provide additional protection against grass pollen-induced nasal symptoms, even in this study set-up where patients were allowed to use their rescue medication if the symptoms were severe and patients had extreme discomfort. However, the observed effect decreased with increasing pollen concentrations. Furthermore, when participants were taking rescue medication because their symptoms were more severe, the baseline protective effect of the probiotic was not sufficient to control disease. The probiotic intervention can thus provide some benefit up to a certain point, beyond which rescue medication may still be necessary (Fig. 4D). Of interest, we observed a considerable time for the probiotics to exert their effects, which started at day 15 and reached a maximal effect 29 days after the start of the trial. A maximal effect size of 2 weeks is also shown for intranasal corticosteroid sprays, which are used as first-line agents in patients with moderate-to-severe and persistent AR (27). In the study by Mårtensson et al., no effects on the self-reported TNSS were found by the probiotic mixture in their nasal allergen challenged model, where they applied the strains for a short timeframe of 3 weeks (26). However, some trials using oral instead of nasal probiotics did report beneficial effects on symptoms. For instance, a 10-week study with fermented milk containing *L. rhamnosus* GG and *L. gasseri* TMC0356 in Japanese seasonal AR patients decreased the mean self-reported symptom score (weekly mean of daily measures via validated questionnaire) for nasal blockage starting from 6 weeks compared to the placebo group, which reached a significant effect after 9 weeks (23). It should be noted, however, that this study did not take pollen concentrations into consideration and calculated weekly means of the daily measures of the symptoms scores, resulting in missing information. Therefore, we preferred the mixed linear model used here taking daily consecutive values into account for TNSS and pollen concentrations, similarly as done in other research related to pollen concentrations and allergy symptoms (28, 29).

In addition to the effects on the symptoms, immunomodulatory properties are of high interest for probiotic application in AR patients. In this study, we found a significant decrease in IL-4 in saliva ($P < 0.05$) and IL-13 in nasal fluid of the probiotic-treated group ($P < 0.0001$) at the end of the intervention. The decrease in IL-13 was also seen in nasal fluid of the placebo group, and while additional research is needed, we hypothesize that one of the reasons for this decrease is that our study set-up allowed the use of rescue medication. No effect on IL-5 was observed in this study. Previous work by Mårtensson et al. with the probiotic nasal spray did not report any effect on cytokine levels in

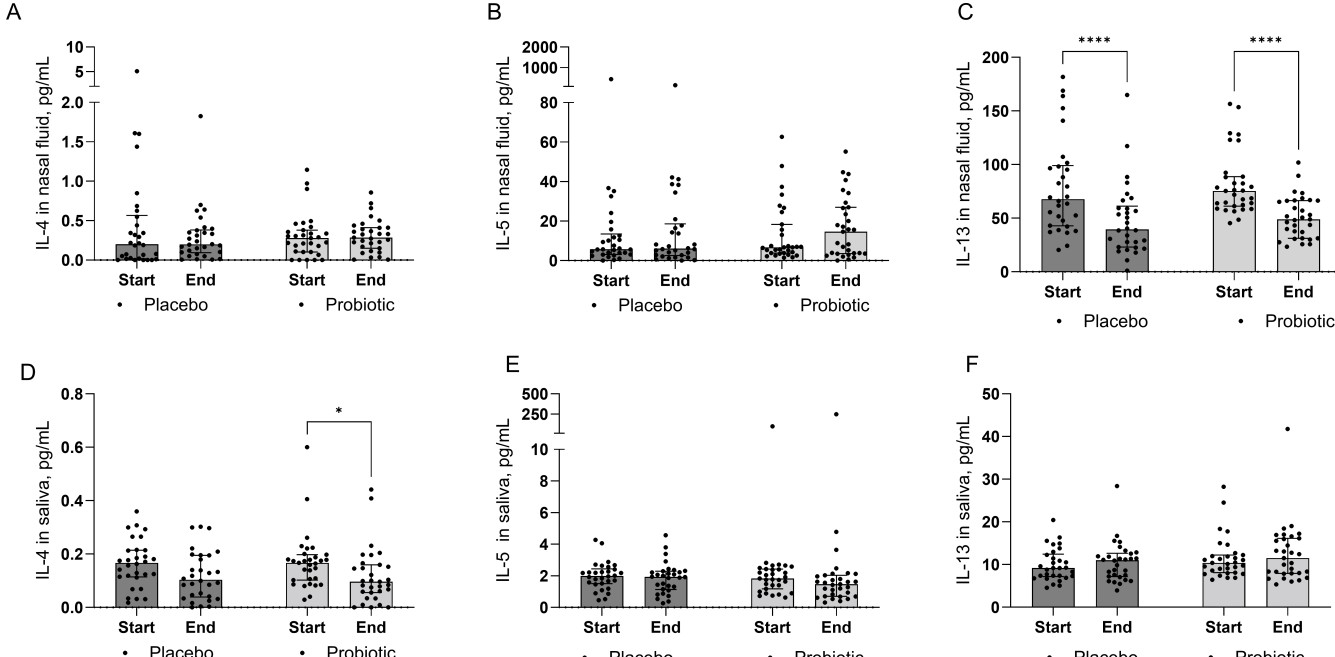

**FIG 5** Cytokine levels in nasal fluid (A–C) and saliva (D–F) of study participants at baseline (start) and 8 weeks (end) of treatment. Pro-allergic cytokines IL-4 (A, D), IL-5 (B, E), and IL-13 (C, F) were measured. Data depicted as median with interquartile range per condition with individual data points per participant; ns $P > 0.05$, *$P < 0.05$, **$P < 0.01$, ***$P < 0.001$, and ****$P < 0.0001$ as determined by a two-way ANOVA followed by Šídák's multiple comparisons test. LGG, *Lacticaseibacillus rhamnosus* GG.

nasal fluid (26). On the other hand, immunomodulatory effects of *L. rhamnosus* GG on Th2 cytokines have been widely described *in vitro* and in murine models. For instance, intranasally administered *L. rhamnosus* GG in mice resulted in decreased IL-13 and IL-5 lung levels (24), and heat-killed *L. rhamnosus* GG is able to inhibit cedar pollen-induced IL-4 and IL-5 production *in vitro* in PBMCs of allergic patients to Japanese cedar pollen (23). The observed decrease in saliva IL-4 and nasal IL-13 levels in our study is particularly noteworthy as such effects are often challenging to observe in human trials.

An important limitation of our work presented here is that measurements for microbiome and immunological biomarkers were only collected at baseline and after the 8 weeks treatment period, when the patients visited the ear-nose-throat (ENT) specialist. Therefore, we might have missed biological changes that could happen within this period. We observed, for instance, based on our mixed-effect linear model, that the probiotic chewables showed a minimal time needed to treat of 2–3 weeks, which might have been an important time point for sampling to observe biological differences. However, increasing sample collection time points at the ENT practice had the risk of reducing patients' compliance to complete the trial. In follow-up research, this can be increased, using the positive indications here to motivate future participants. With enrollees' compliance in mind, we also used mainly self-reported TNSS values for the other time points when the patients were not visiting the ENT specialist. Another limitation of our study was that, despite randomization, participants in the probiotic group more frequently used corticosteroid sprays at baseline (46.7%) compared to the placebo group (23.3%). This indicates that the probiotic group might have experienced more severe complaints at baseline. In order to account for this effect, the use of corticosteroids at the start of the study was implemented in the mixed-effect linear model. Most of our study population was also patients with moderate to severe, persistent symptoms (based on ARIA guidelines), while patients with mild symptoms could benefit more from this therapy. Finally, our patients were allowed to use rescue medication (i.e., their standard treatment) because we evaluated a food supplement and

not a drug. Our study was the first in human clinical trial with this specific supplement for this indication: without previous data for allergy patients, the risk to impact their QOL was considered too high. Therefore, we allowed the option to take rescue medication, but only when symptoms were not controlled. The need for rescue medication depended on the QOL perception of the patients. This could have impacted clear interpretations of certain effects or trends observed in this study, but was considered a valuable approach respecting the QOL of the patients. Nevertheless, we took the rescue medication and history into account in the model.

Despite these limitations, our study provides new insights for the development of URT-targeting probiotics for AR patients and related patient groups: trafficking to the nose is possible if a slow-release formulation is chosen. In addition, this can result in measurable benefits for the patients. Our data also clearly show that a minimal treatment time of approximately two to three weeks is necessary. Probiotic/microbiome therapeutics that target the airways are thus a promising strategy for AR, alone or in combination with existing therapies, depending on the severity of the symptoms. Thus, nasal targeting probiotic/microbiome therapy warrants further investigation in respiratory allergies, especially in preventive settings and as an add-on strategy to the existing medication options.

## MATERIALS AND METHODS

### Clinical trial design

A double-blind, placebo-controlled clinical trial was performed with a probiotic chewable in seasonal AR patients. The trial was designed as a semi-preventive set-up, with the start of the intervention approximately four weeks before the start of the grass pollen season and four weeks during the season. The estimation of the start of the season was done based on the airborne pollen concentration data from previous years. Approval was obtained from the committee of medical ethics (Antwerp University Hospital/UAntwerpen, B3002020000086) on 8 June 2020 by the local institutional review board. The actual intervention trial was conducted from 27 April 2021, to 4 August 2021, at the ENT Department of the Antwerp University Hospital and ENT Kalmthout. Patients started the intervention between 27 April and 25 May 2021. The trial was registered on ClinicalTrials.gov (NCT04898686) and conducted according to good clinical practice. An informed consent was obtained from all participants prior to inclusion. Data on mean daily pollen concentrations (pollen grains/m$^3$) of the taxon grasses (family Poaceae), measured by the Hirst method at the Brussels station of the Belgian aerobiological surveillance network, were provided by Sciensano (Belgium) (Suppl. Excel file for raw data). This station is geographically the closest aerobiological station to Antwerp. Furthermore, it has been shown that temporal variations in annual pollen levels almost always follow a similar trend in the Benelux (30).

### Participants

Seasonal AR adult patients (>18 y/o) were recruited via the Antwerp University Hospital, ENT medical practice Kalmthout, or via social media/website of the Lab of Applied Microbiology and Biotechnology (University of Antwerp). Patients were invited for an intake consult and SPT to evaluate whether all inclusion criteria were met. Sensitization to other major allergens was an exclusion criterion, which was evaluated for house dust mite, grass pollen, birch pollen, mugwort pollen, plantain pollen, mold mixture, and cat, dog, and horse allergens. Patients were allowed to use rescue medication in case the symptoms were not controlled for ethical reasons. ARIA (Allergic Rhinitis and its Impact on Asthma) guidelines were used to subdivide the patient population according to their symptoms and severity (31). Two weeks prior to study start, participants were not allowed to take other probiotics, neither during the trial. In addition, the use of antibiotics before or during the trial was an exclusion criterion. Participants filled in

a general questionnaire at the start of the trial, where additional questions on antibiotic use and probiotic/fermented foods were asked (see supplementary information) to evaluate compliance with these guidelines.

## Randomization and masking

Randomization occurred in blocks of six patients with stratification for symptoms (intermittent or persistent based on ARIA guidelines) and severity (mild or moderate/severe based on ARIA guidelines), using a randomization list generated with the Sealed Envelope web service (https://www.sealedenvelope.com/), by the study coordinator. All other researchers and doctors involved in the study were blinded. Participants were enrolled and assigned to study groups by the study coordinator at the time of the first visit. Probiotic and placebo products were given the label A or B. Participants, investigators, and outcome assessors were blinded for the treatment allocation until all analyses were performed and the study groups were unblinded by the formulation team.

## Intervention

Probiotic and placebo chewables were supplied by DSM iHealth (USA) and were indistinguishable in taste, form, and color. The probiotic chewables consisted of *L. rhamnosus* GG ($10^{10}$ CFUs per tablet) with xylitol, microcrystalline cellulose, stearic acid, natural orange flavor, silica, magnesium stearate, citric acid, and malic acid. Placebo chewables had the same composition without the bacterial component. Quality control was performed to make sure no cross-contamination could occur between the placebo and probiotic chewables.

## Study procedures

Patients were asked to use the probiotic or placebo chewables for 8 weeks, once daily with the instructions not to eat or drink within 1 h after intake, and to fill in an online diary via a survey platform (Qualtrics) for each day of the study reporting usage of the chewable, allergy-related symptoms, and medication use. After the intervention period, a 2-week follow-up period was included where participants were asked to fill in the online diary. Allergy complaints were evaluated via the TNSS (32). This is a validated score that is measured based on four different symptoms: (i) blocked nose, (ii) runny nose, (iii) itching nose, and (iv) sneezing.

At the start of the study and after the 8-week intervention group, participants had a visit at the ENT Department of the Antwerp University Hospital or ENT Kalmthout with the responsible ENT specialist and study coordinator. At each visit, the following samples were collected: two nasopharyngeal swabs (Copan, FLOQSwabs 501CS01), one oropharyngeal swab (Copan, FLOQSwabs 503CS01), saliva, and nasal fluid (Merocel). All samples were pseudonymized and registered in the in-house biobank decentralized hub to comply with the most recent GDPR regulations in Belgium on biobanking human samples (KB 2018/30209).

## Outcomes

The primary clinical outcomes of this trial were (i) the transfer of *L. rhamnosus* GG from the chewable to the oronasopharyngeal region, assessed via sequencing and qPCR, and (ii) changes in the score of AR symptoms, assessed via the TNSS. The secondary study outcomes included: (i) changes in the microbiome of the oronasopharyngeal region, (ii) changes in absolute numbers of specific airway pathogens, (iii) frequency of medication use, (iv) changes in local cytokine levels in nasal fluid and saliva, and (v) changes in general nose and mouth health.

Finally, the correlation of the TNSS and microbiome with the pollen concentrations was included as explorative (post hoc) analysis.

## Sample size

A sample size calculation was performed based on the main research question: the transfer of *L. rhamnosus* GG from the chewable to the oronasopharyngeal region. A sample size of 42 subjects allows detecting a 40% difference in the estimated colony-forming units (CFU) counts before and after intervention, a power of 90%, and a type 1 error of 0.05. With 64 subjects included in the analysis, a sufficient sample size was reached while taking possible loss to follow-ups into account. Sample size calculations were done using the "WMWssp" package (33) (publicly available at http://github.com/happma/WMWssp) in R version 4.3.1 (34) .

## Bacterial DNA extraction from oropharyngeal and nasopharyngeal swabs and Illumina MiSeq 16S rRNA amplicon sequencing

Oro- and nasopharyngeal swabs were stored at −20°C until further processing. Prior to DNA extraction, all samples were vortexed 15–30 s and 500 µL of the eNAT buffer was used for automatic extraction using PowerSoil Pro Ht Kit (QIAcube HT 9001794). Negative extraction controls were included at regular time points throughout the study and used for our quality control pipeline. DNA concentrations were measured using the Qubit 3.0 Fluorometer (Life Technologies, Ledeberg, Belgium).

Illumina MiSeq 16S rRNA gene amplicon sequencing was performed on the extracted DNA from the swabs to investigate the bacterial communities. An in-house optimized protocol was followed, as described (35). Processing and quality control of the reads was performed using the R package DADA2, version 1.6.0. All data handling and visualization were performed in R version 3.4.4 (R Core Team 2018) using the tidyverse set of packages and the in-house package tidyamplicons, version 0.2.1 (publicly available at github.com/SWittouck/tidyamplicons).

## Detection of the presence of *L. rhamnosus* using qPCR

In-house specific primers for a subset of *L. rhamnosus* strains, including *L. rhamnosus* GG, were developed based on the pangenome and used for a qPCR assay (Table 2), as described in Eilers et al. (36). Potential off-targets were analyzed using BLAST, optimized for smaller fragments, confirming specificity for *L. rhamnosus*.

Each qPCR reaction consisted of 4 µL of each extracted DNA sample, 10 µL Power SYBR Green PCR Master Mix, 0.3 µL of each primer (20 µM), and 5.4 µL of RNase-free water. The cycle threshold (Ct) value of each sample was used to calculate the concentration of the strain present in the sample based on a standard curve. Non-template controls were included for each run.

## Processing of nasal fluid and saliva samples

Nasal tampons (Ivalon) were weighed before and after sample collection. Sterile saline solution (0.9% NaCl) was added to the tampons equal to 4 x the volume/weight of the nasal fluid. The sponges with saline were incubated for 1 h at 4°C to soak. Subsequently, the liquid was squeezed out of the sponges by placing them in a sterile 5 mL syringe, and the syringe was additionally centrifuged at 1,500 × *g* for 5 min at 4°C. The supernatant representing nasal fluid was stored at −80°C. Saliva samples collected via a swab were immediately placed in 0.5 mL sterile phosphate-buffered saline. Subsequently, the tube was vortexed, and the swab was placed in a syringe that was centrifuged at 1,500 × *g* for 5 min, 4°C, to collect the remaining fluid in the same tube with saliva diluted in saline. The supernatant representing saliva in saline was stored at −80°C. Before cytokine measurements, the samples were thoroughly vortexed.

## MSD assay on nasal fluid and saliva

The cytokine levels in nasal fluid and saliva samples processed as described above were analyzed using a custom multiplex assay (Meso Scale Discovery U-PLEX TH1/TH2 Combo

human assay). Briefly, after coating the 96-well plates with a multiplex coating solution containing U-PLEX linker-coupled antibodies for 1 h, 25 µL of sample or calibrator combined with 25 µL of diluent was added in each well and incubated for 1 h. Afterward, the detection antibody solution was added to the plate for 1 h. In between each step, the plate was washed three times with PBS‐Tween (0.05%). Finally, after the addition of the MSD Gold Read Buffer, the plate was analyzed on the QuickPlex SQ 120 (MSD) instrument (Rockville, MD, USA).

## Statistical analysis

Per-protocol analysis was performed on participants who completed the study and provided samples at all time points. To assess differences between the estimated CFU counts before and after intervention and between treatment groups, normality was rejected using histograms and Shapiro-Wilk's test. Afterward, nonparametric Mann-Whitney tests were used in R.

To assess differences between treatment groups at different time points for the tested cytokine levels, Two-way ANOVA followed by Šídák's multiple comparisons test was used in the GraphPad Prism software version 9.2.0. Besides treatment, we have also evaluated time (start or end of study), participant, and interaction between time and treatment as factors that could serve as sources of variation in the Two-way ANOVA.

The TNSS was processed prior to analysis. Missing values were imputed by taking the next non-missing TNSS value. Following this, the signal was smoothed, for each day taking the average TNSS of the 3 days before and after. Finally, as the scores are subjective intra-individual measurements, the TNSS was standardized to standard normal within each participant. Rescue medication intake was imputed in the same way as the TNSS.

The effect of treatment and pollen concentrations on the TNSS was tested with a mixed effect model using the R lmerTest package: TNSS ~ Treatment × Pollen × rescue + corticosteroid history + (1 + rescue|participant) + (1|startgroup). We investigated the interaction between treatment group, pollen concentration, and rescue medication intake. To account for a history of corticosteroid use, this was included in the model. Random effects were included to account for the personal effects of rescue medication, starting group, and intra-individual effects. We do not include time in this model, as we do not expect an effect of time *per se*. Rather, we expect a different effect of the pollen concentration on the TNSS in the two study conditions, which we formally test in the interaction effect. To investigate the minimum time to treat, we tested the model on subsets where the first *n* days have been removed. We tested the model for *n* = 0.49, where the first, up to the first 49 days had been removed. We adjusted for multiple testing with the Benjamini–Hochberg procedure with an alpha of 0.05. Microbiome associations were tested with Multidiffabundance (37), making use of the alpha diversity test (Shannon diversity) and Maaslin2 (38). We evaluated the association between the microbiome and the probiotic treatment effect (~time point POAC + corticosteroid_history +dna_pcr_reads + start_group + (1|participant), evaluated only within the probiotic treatment group), and the TNSS (~TNSS + POAC + corticosteroid_history + dna_pcr_reads + start_group + (1|participant)). We adjusted for multiple testing with the Benjamini–Hochberg procedure with an alpha of 0.05.

## ACKNOWLEDGMENTS

I.D.B., I.S., and K.M. were supported by grants from Research Foundation - Flanders (FWO postdoctoral grants 12S4222N, 1277222N, and 12Z0622N). E.C. is supported by iBOF grant POSSIBL. S.L., T.G., and T.E. were supported by the European Research Council grant (Lacto-Be 852600). Airborne pollen concentration data collection was co-financed by the Brussels Environment agency and the Flemish agency for Care and Health.

We would like to thank the Lebeerlab team and all study participants that participated in this trial. We would also like to thank the nurses at the ENT department of the UZA for their help with the skin prick tests and blood collection, in particular, Heidi

Hoogewijs. Finally, we would like to thank Biobank Antwerpen (Antwerp, Belgium; ID: BE 71030031000).

I.D.B.: Conceptualization, Data collection, Data curation, Formal analysis, Methodology, Visualization, Original draft, Writing—review and editing. I.S.: Conceptualization, Data collection, Data curation, Formal analysis, Methodology, Visualization, Original draft, Writing—review and editing. S.L.: Conceptualization, Original draft, Resources, Writing—review and editing. O.M.V.: Conceptualization, Resources. A.V.: Conceptualization, Data collection. K.V.G.: Conceptualization, Data collection, Resources. E.C.: Data collection, Data curation, Formal analysis, Methodology, Writing—review and editing. K.V.d.B.: Data collection, Methodology, Writing—review and editing. N.B.: Data collection, Writing—review and editing. A.P.: Data collection, Writing—review and editing. T.E.: Data curation, Formal analysis, Methodology, Visualization, Writing—review and editing. T.G.: Data curation, Formal analysis, Methodology, Visualization, Writing—review and editing. K.M.: Methodology, Writing—review and editing. S.C.-C.: Methodology, Writing—review and editing. F.D.W.: Methodology, Writing—review and editing. S.K.-S.: Resources, Writing—review and editing. P.W.H.: Resources, Writing—review and editing.

## AUTHOR AFFILIATIONS

[1]Laboratory of Applied Microbiology and Biotechnology, Department of Bioscience Engineering, University of Antwerp, Antwerp, Belgium

[2]Faculty of Medicine and Health Sciences, University of Antwerp, Wilrijk, Belgium

[3]ENT, Head and Neck Surgery, Antwerp University Hospital, Edegem, Belgium

[4]Department of Microbiology, Immunology and Transplantation, Allergy and Clinical Immunology Research Group, KU Leuven, Leuven, Belgium

[5]Data Science Institute and I-BioStat, Hasselt University, Diepenbeek, Belgium

[6]Molecular Pathology Group, Laboratory of Cell Biology & Histology, Faculty of Medicine and Health Sciences, University of Antwerp, Wilrijk, Belgium

[7]Service Mycology & Aerobiology, Sciensano, Brussels, Belgium

[8]Clinical Division of Ear, Nose and Throat Disease, University Hospitals Leuven, Leuven, Belgium

[9]ENT Kalmthout, Kalmthout, Belgium

## AUTHOR ORCIDs

Ilke De Boeck ⓘ http://orcid.org/0000-0003-0764-8515
Irina Spacova ⓘ http://orcid.org/0000-0003-0562-7489
Sarah Lebeer ⓘ http://orcid.org/0000-0002-9400-6918

## FUNDING

| Funder | Grant(s) | Author(s) |
| --- | --- | --- |
| Fonds Wetenschappelijk Onderzoek | 12S4222N | Ilke De Boeck |
| Fonds Wetenschappelijk Onderzoek | 1277222N | Irina Spacova |
| Fonds Wetenschappelijk Onderzoek | 12Z0622N | Katleen Martens |
| iBOF | POSSIBL | Eline Cauwenberghs |
| European Research Council | 852600 | Tom Eilers |
| | | Thies Gehrmann |
| | | Sarah Lebeer |

## AUTHOR CONTRIBUTIONS

Ilke De Boeck, Conceptualization, Data curation, Formal analysis, Methodology, Visualization, Writing – original draft | Irina Spacova, Conceptualization, Data curation, Formal analysis, Methodology, Visualization, Writing – original draft, Writing – review

and editing | Eline Cauwenberghs, Data curation, Formal analysis, Methodology, Writing – review and editing | Tom Eilers, Data curation, Formal analysis, Methodology, Visualization, Writing – review and editing | Thies Gehrmann, Data curation, Formal analysis, Methodology, Visualization, Writing – review and editing | Karlien Van den Bossche, Methodology, Writing – review and editing | Katleen Martens, Methodology, Writing – review and editing | Sandra Condori-Catachura, Methodology, Writing – review and editing | Kato Michiels, Methodology, Writing – review and editing | Fien De Winter, Methodology, Writing – review and editing | Samir Kumar-Singh, Resources, Writing – review and editing | Nicolas Bruffaerts, Data curation, Writing – review and editing | Ann Packeu, Data curation, Writing – review and editing | Peter W. Hellings, Resources, Writing – review and editing | Anneclaire Vroegop, Conceptualization, Data curation, Writing – review and editing | Klara Van Gool, Conceptualization, Resources, Writing – review and editing | Olivier M. Vanderveken, Conceptualization, Resources, Writing – review and editing | Sarah Lebeer, Conceptualization, Resources, Writing – original draft

## DATA AVAILABILITY

All data produced in the present study are available upon reasonable request to the authors. The sequencing data were deposited in the ENA under accession number PRJEB52101.

## ADDITIONAL FILES

The following material is available online.

### Supplemental Material

**Supplemental figures, table, and questionnaire (Spectrum00773-25-S0001.docx).** Fig. S1 to S3, Table S1, and questionnaire.

### Open Peer Review

**PEER REVIEW HISTORY (review-history.pdf).** An accounting of the reviewer comments and feedback.

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
