## [Reviewer comments · Microbiology Spectrum]

Microbiology Spectrum

Lacticaseibacillus rhamnosus GG in a chewable colonizes the nose and facilitates local immune benefits in allergic rhinoconjunctivitis patients

Ilke De Boeck, Irina Spacova, Eline Cauwenberghs, Tom Eilers, Thies Gehrman, Karlien Van den Bossche, Katleen Martens, Sandra Condori-Catachura, Kato Michiels, Fien De Winter, Samir Kumar-Singh, Nicolas Bruffaerts, Ann Packeu, Peter Hellings, Anneclaire Vroegop, Klara Van Gool, Olivier Vanderveken, and Sarah Lebeer

Corresponding Author(s): Ilke De Boeck, Universiteit Antwerpen

Review Timeline:

Submission Date:	March 24, 2025
Editorial Decision:	May 6, 2025
Revision Received:	May 27, 2025
Accepted:	June 27, 2025

Editor: Hao-Yu Liu

Reviewer(s): The reviewers have opted to remain anonymous.

Transaction Report:

DOI: <https://doi.org/10.1128/spectrum.00773-25>

Re: Spectrum00773-25 (Lacticaseibacillus rhamnosus GG in a chewable colonizes the nose and facilitates local immune benefits in allergic rhinoconjunctivitis patients)

Dear Dr. Ilke De Boeck:

Thank you for the privilege of reviewing your work. Below you will find my comments, instructions from the Spectrum editorial office, and the reviewer comments.

Revision Guidelines

Sincerely,
Hao-Yu Liu
Editor
Microbiology Spectrum

Reviewer #1 (Comments for the Author):

As far as I can see, this manuscript has already been reviewed once from two reviewers and updated according to the input provided in the first review round. I think that these reviewers provided valid points but also that the revisions made has been fair and helped to address the input from these review questions.

I have not reviewed this manuscript previously though and I find this study interesting in general, but there are still remaining

questions that still need to be addressed:

General comments:

The description of the statistical analysis in the method section still needs revision. First, did you check for the data distribution, e.g. normality of the data and how did you handle this? I also think that the Two-way ANOVA need to be described more, for example, what was the different factors evaluated beside treatment groups? Which software did you use? This is not very clear. Moreover, the description in line 431-438 is also a bit unclear, mainly in terms of where these results are presented in the results section.

In line 387-394 you describe detection of the probiotic strain using qPCR. It seems like the primers were designed in the present study. If so, what kind of validation regarding the specificity of these primers were performed? For example to prevent unspecific binding of primers to other related strains. This is not described at all.

It seem that you also detected the probiotic strain in the control group, although at a lower level compared with the probiotic group. And it seem also that this occurred in the end sampling. This is just mentioned in the results, but it also need to be mentioned and brought up in the discussion.

Did you have any knowledge whether the participants use or had previously used probiotics or/and antibiotics prior to the intervention (at least in close proximity to the interventions would be relevant to know)? This information would also be quite relevant in this study.

Sometimes you write verum, sometimes you used probiotic. To me the term verum is not very clear, so I prefer probiotic.

Ensure that you can read and interpret the figure by looking at the figure and figure legend. For example by writing out abbreviations and explanation of what different colors mean. Not always done.

Both Figure 4 and its legend is a bit hard to understand. No description of panel D in the legend. The figure contains panel A-D, but in panel D you also use symbol A and B, this is confusing. Also no explanation for the different colors in panel D. In line 488 you describe (A) probiotic treatment and (B) pollen concentration. I assume you refer to the different panels? If so, then it should be (A) for pollen and (B) for Probiotic treatment.

Specific comments

In table 1, you need to define abbreviations (TNSS and SNOT-22). Can be done as a foot note.

Lines 157-160. Results from the adonis not significant I assume then? Could be added in the text.

Line 271: Define ENT (Defined in line 312, but should be defined first time it appear in the text).

Line 350: Use abbreviation TNSS.

Line 379: What did you do with the extraction controls? Did you include them in sequencing?

Line 438-439: You write that you adjusted multiple testing using B-H procedure, but unclear which statistical tests? In most parts you refer to Sidaks multiple comparison test.

Line 476: Remove abbreviations NF and OF. These are not used in the figures, and thus no need to mention the abbreviation in the legend.

In figure 2, you use verum. To me clearer if you used probiotics instead.

Figure 3 has low resolution, thus it is hard to see text in the figure, so try to may it sharper.

Line 481. Add "black vertical lines" to clarify that it is these lines you refer to.

Reviewer #1 (Comments for the Author):

As far as I can see, this manuscript has already been reviewed once from two reviewers and updated according to the input provided in the first review round. I think that these reviewers provided valid points but also that the revisions made has been fair and helped to address the input from these review questions.

I have not reviewed this manuscript previously though and I find this study interesting in general, but there are still remaining questions that still need to be addressed:

We would like to thank the reviewer for the time to review our manuscript and the suggestions for improvement. We have provided a point-by-point response and indicated our adjustments with track changes. The lines indicated below correspond to the manuscript file with track changes.

General comments:

The description of the statistical analysis in the method section still needs revision. First, did you check for the data distribution, e.g. normality of the data and how did you handle this? I also think that the Two-way ANOVA need to be described more, for example, what was the different factors evaluated beside treatment groups? Which software did you use? This is not very clear. Moreover, the description in line 431-438 is also a bit unclear, mainly in terms of where these results are presented in the results section.

Answer: We thank the reviewer for this observation. More information on the methods has been added in lines 435-442, including the GraphPad Prism software used for the Two-way ANOVA and the factors that have been included in the analysis. For the estimated colony forming units from qPCR, normality was evaluated in R visually and using Shapiro-Wilk's test before continuing with a nonparametric test. We also elaborate more on the Two-way ANOVA in an additional sentence in the results section lines 209-212 of the Results section. We acknowledge that ANOVA assumes normal distribution; however, this assumption is known to be robust to moderate deviations from normality. Considering our goal to test the effects of multiple factors, non-parametric options to do this are statistically more limited and less commonly supported. Our design specifically includes comparable group sizes and sample sizes of much more than 20 participants per group, which helps ensure the robustness of ANOVA results.

I. 209-212: Two-way ANOVA analysis identified time (start or end of study) as a significant source of variation for the levels of IL-4 in saliva and IL-13 in nasal fluid, while individual differences between participants were a significant source of variation for IL-4, IL-5 and IL-13 in saliva, and IL-13 in nasal fluid.

I. 435-442: To assess differences between the estimated CFU counts before and after intervention and between treatment groups, normality was rejected using histograms and Shapiro-Wilk's test. Afterwards, nonparametric Mann-Whitney tests were used in R. To assess differences between treatment groups at different time points for the tested cytokine levels, Two-way ANOVA followed by Šídák's multiple comparisons

test was used in the GraphPad Prism software version 9.2.0. Besides treatment, we have also evaluated time (start or end of study), participant and interaction between time and treatment as factors that could serve as sources of variation in the Two-way ANOVA.

In line 387-394 you describe detection of the probiotic strain using qPCR. It seems like the primers were designed in the present study. If so, what kind of validation regarding the specificity of these primers were performed? For example to prevent unspecific binding of primers to other related strains. This is not described at all.

Answer: We apologize because this is indeed missing in our manuscript. The exact method is described in our preprint: <https://doi.org/10.21203/rs.3.rs-4182624/v1>

This is now included in our methods section on lines 405-409:

In-house specific primers for a subset of *L. rhamnosus* strains, including *Lacticaseibacillus rhamnosus* GG were developed based on the pangenome and used for a qPCR assay (Table 2), as described in Eilers et al. Potential off-targets were analyzed using BLAST, optimized for smaller fragments, confirming specificity for *Lacticaseibacillus rhamnosus*.

It seem that you also detected the probiotic strain in the control group, although at a lower level compared with the probiotic group. And it seem also that this occurred in the end sampling. This is just mentioned in the results, but it also need to be mentioned and brought up in the discussion.

Answer: This is now added in the discussion on lines 225-231

Of note, after the 8 week intervention, some of the participants in the control group were also positive for *L. rhamnosus* GG via qPCR, but this was significantly lower compared to the probiotic group ($p = 0.02$ for nasopharynx and $p < 0.0001$ for oropharynx). Possible explanations include its presence as a low-abundance member of the normal nasal microbiota, as previously reported (De Boeck et al. 2022). Additionally, transfer through the consumption of other fermented or dairy products cannot be excluded. Lastly, the potential for low-level cross-contamination during sample processing or sequencing should also be considered.

Did you have any knowledge whether the participants use or had previously used probiotics or/and antibiotics prior to the intervention (at least in close proximity to the interventions would be relevant to know)? This information would also be quite relevant in this study.

Answer: This is a very valuable question from the reviewer, and we have this data available both on antibiotics and probiotic use via a questionnaire, which we will add as supplementary information. Furthermore, our inclusion criteria stated the following:

- **No antibiotic use at start or during the trial**

- No use of probiotics in the past 2 weeks before start of the trial or during the trial

This is now specified in the materials and methods section on lines 341-346

Two weeks prior to study start, participants were not allowed to take other probiotics, neither during the trial. In addition, the use of antibiotics before or during the trial was an exclusion criterium. Participants filled in a general questionnaire at the start of the trial, where additional questions on antibiotic use and probiotic/fermented foods were asked (see supplementary information) to evaluate compliance with these guidelines.

Sometimes you write verum, sometimes you used probiotic. To me the term verum is not very clear, so I prefer probiotic.

Answer: We have changed verum to probiotic throughout the manuscript and in the figures.

Ensure that you can read and interpret the figure by looking at the figure and figure legend. For example by writing out abbreviations and explanation of what different colors mean. Not always done.

Both Figure 4 and its legend is a bit hard to understand. No description of panel D in the legend. The figure contains panel A-D, but in panel D you also use symbol A and B, this is confusing. Also no explanation for the different colors in panel D. In line 488 you describe (A) probiotic treatment and (B) pollen concentration. I assume you refer to the different panels? If so, then it should be (A) for pollen and (B) for Probiotic treatment.

Answer: we have added additional information in our figure legends for clarity. We apologize for Figure 4, because we did not properly adjust our legend after we made a new figure. We have now adjusted the legend. All changes to the figure legends can be found starting from l. 490 in the manuscript. Specifically for Figure 4, the legend now reads:

Figure 4. Impact of treatment on TNSS in relation to grass pollen exposure and medication use in terms of effect size. A-D: Mixed effect linear model considering treatment, pollen concentrations, rescue medication intake, and their interactions to investigate the effect on the TNSS. When considering the first x days as a minimum time to treat period (plotted on the x-axis), the effect sizes from (A) pollen concentration (POAC; family Poaceae, grasses), (B) probiotic treatment, and (C) rescue medication are plotted on the y-axis. Significant values are plotted in green, insignificant values in pink. Negative effect size reflects a lower TNSS. The black points indicate the selected time-point where effect size is at its maximum for the model, which is an indication for the minimal time to treat. Panel D shows linear regression at day 29 threshold which has maximum effect for probiotic treatment, showing the interaction of TNSS and pollen counts for both group with (dotted lines) and without (fixed lines) rescue medication.

Specific comments

Answer: We thank the reviewer for the careful review and have adapted all comments. In table 1, you need to define abbreviations (TNSS and SNOT-22). Can be done as a foot note. **The abbreviations are added.**

Lines 157-160. Results from the adonis not significant I assume then? Could be added in the text. **This is included.**

Line 271: Define ENT (Defined in line 312, but should be defined first time it appear in the text). **This is included.**

Line 350: Use abbreviation TNSS. **We have adapted this.**

Line 379: What did you do with the extraction controls? Did you include them in sequencing?

Answer: yes they were included in our quality control. Bacterial taxa that were also present in high abundances in our negative extraction controls were excluded from the dataset. The same was done for PCR controls. This is now clarified in the text (l. 396-397).

Line 438-439: You write that you adjusted multiple testing using B-H procedure, but unclear which statistical tests? In most parts you refer to Sidaks multiple comparison test.

Answer: For the cytokine data, we have indeed implemented the Šidák multiple comparison test in GraphPad, which is in itself a statistical test that already includes correction for multiple comparisons. In addition, we have now added in the manuscript that the Benjamini-Hochberg procedure was used for the mixed-effect linear model (l. 458-459).

Line 476: Remove abbreviations NF and OF. These are not used in the figures, and thus no need to mention the abbreviation in the legend. **We have removed these abbreviations.**

In figure 2, you use verum. To me clearer if you used probiotics instead. **This is adapted.**

Figure 3 has low resolution, thus it is hard to see text in the figure, so try to may it sharper. **We have included another figure.**

Line 481. Add "black vertical lines" to clarify that it is these lines you refer to. **We have adjusted the figure caption.**

Re: Spectrum00773-25R1 (Lacticaseibacillus rhamnosus GG in a chewable colonizes the nose and facilitates local immune benefits in allergic rhinoconjunctivitis patients)

Dear Dr. Ilke De Boeck:

Your manuscript has been accepted, and I am forwarding it to the ASM production staff for publication. Your paper will first be checked to make sure all elements meet the technical requirements. ASM staff will contact you if anything needs to be revised before copyediting and production can begin. Otherwise, you will be notified when your proofs are ready to be viewed.

Sincerely,
Hao-Yu Liu
Editor
Microbiology Spectrum